# Ethanol Extract of Adlay Hulls Suppresses Acute Myeloid Leukemia Cell Proliferation via PI3K/Akt Pathway Inhibition

**DOI:** 10.3390/cimb47050358

**Published:** 2025-05-13

**Authors:** Guangjie Li, Wenyuan Yang, Jiahui Xu, Ziqian Liu, Zhijian Li, Xiaoqiu Wu, Tongtong Li, Ruoxian Wang, Yamin Zhu, Ning Liu

**Affiliations:** 1International Research Centre for Food and Health, College of Food Science and Technology, Shanghai Ocean University, Shanghai 201306, China; guangjieli7757@163.com (G.L.); 18830396365@163.com (J.X.); liu873075580@163.com (Z.L.); lizhij1999@163.com (Z.L.); xiaoqiu8813@163.com (X.W.); prayer27t@163.com (T.L.); 15226862577@163.com (R.W.); 2School of Life Sciences, Henan University, Kaifeng 475003, China; wy.yang@henu.edu.cn; 3Marine Biomedical Science and Technology Innovation Platform of Lin-Gang Special Area, Shanghai 201306, China; 4Department of Chemistry, College of Food Science and Technology, Shanghai Ocean University, Shanghai 201306, China; 5Shanghai Engineering Research Center of Aquatic-Product Processing and Preservation, Shanghai 201306, China

**Keywords:** acute myeloid leukemia, adlay hull extract, proliferation, apoptosis, bioactive components, PI3K/Akt pathway

## Abstract

Acute myeloid leukemia (AML) is a common hematologic malignancy in the elderly with frequent relapse and poor prognosis. Limited treatments highlight the need for novel natural anticancer compounds. Adlay, valued for its medicinal and dietary properties, exhibits anti-inflammatory and anticancer effects. However, research on adlay hulls, particularly their anti-AML bioactive molecules, remains insufficient. This study evaluated the effects of adlay hull ethanol extract (AHE) on AML cell proliferation and apoptosis. AHE was extracted with ethanol and fractionated using n-hexane, ethyl acetate, and n-butanol, followed by silica gel chromatography. Cytotoxicity was assessed via the CCK-8 assay, and mechanisms were analyzed by flow cytometry and Western blotting. The bioactive components were characterized by UPLC-IMS-QTOF-MS. AHE-EA-C (ethyl acetate fraction C) inhibited AML cell proliferation, induced G0/G1 phase arrest, and promoted apoptosis. It suppressed the PI3K/Akt pathway by reducing PI3K and Akt phosphorylation. Using UPLC-IMS-QTOF-MS analysis, a total of 52 compounds with potential anti-AML activity were identified in AHE-EA-C, among which neohesperidin and cycloartanol have been previously reported to exhibit anti-AML activity and thus hold promise as candidates for further development as AML inhibitors. This study is the first to identify adlay hull bioactive components and their anti-AML mechanisms via PI3K/Akt pathway inhibition, providing a foundation for developing natural anti-AML therapies.

## 1. Introduction

Acute myeloid leukemia (AML) is a hematologic malignancy characterized by the clonal expansion of myeloid blasts, impaired differentiation, and disrupted normal hematopoiesis. While AML can affect individuals across all age groups, it is particularly prevalent in older populations, with more than two-thirds of cases diagnosed in patients aged 55 or older [1]. AML is responsible for over 80,000 deaths annually worldwide, and this figure is projected to double over the next two decades [2]. Until recently, treatment options for AML patients were largely restricted to intensive chemotherapy with cytarabine and anthracycline (7 + 3 regimen) or the use of hypomethylating agents [3]. Although 70% to 80% of AML patients under the age of 60 achieve complete remission, the majority eventually relapse [4]. Therefore, the identification of novel natural compounds exhibiting anticancer properties could offer a more effective approach for treating AML.

Given the high toxicity of chemotherapeutic agents and the widespread emergence of resistance, the development of specific molecules targeting key signal transduction pathways is a potentially promising new therapeutic strategy [5]. The PI3K/Akt pathway regulates various cellular processes, such as protein synthesis, cell cycle progression, survival, apoptosis, angiogenesis, and drug resistance [6]. Activation of PI3K/Akt is observed in over 60% of AML patients and is linked to reduced overall survival [5]. PI3K exists in several classes, with class I PI3K being the primary subtype involved in the PI3K/Akt/mTOR signaling network [7]. The mammalian PI3K family consists of several isoforms across different classes. Class I PI3Ks, further categorized into IA and IB, produce 3-phosphoinositide lipids that activate various signaling pathways. Class IA PI3Ks are frequently activated in various cancers, leading to the development and approval of multiple pathway inhibitors [8,9]. For instance, the selective inhibitor of the class I PI3K isoform PI3Kδ, idelalisib, has demonstrated significant therapeutic efficacy in certain hematologic malignancies. However, the therapeutic potential of PI3Kδ inhibitors in AML remains under investigation [10]. Therefore, further in-depth research on PI3K/Akt pathway inhibitors targeting AML-specific molecular aberrations is still needed to develop more effective therapeutic strategies.

Natural products, offering immense structural and chemical diversity unmatched by any synthetic small molecule library, have played a crucial role in drug discovery and remain the best source of drugs and drug leads, with many of their derivatives successfully used in clinical treatments across nearly all therapeutic areas [11,12]. Among these natural resources, adlay (soft-shelled job’s tears, *Coix lachryma-jobi* L. var. *mayuen* Stapf) has emerged as a promising candidate. Belonging to the Poaceae family and the genus *Coix*, adlay produces pear-shaped seeds with shiny, dark brown to gray-black husks. While the seed is an important food source in certain regions of Asia, its husks (adlay hulls) exhibit various medicinal properties, such as anti-inflammatory effects. Epidemiological studies have shown that individuals in some rural areas of Asia who incorporate adlay seeds into their diet tend to have a significantly lower incidence of cancer [13]. In recent years, several researchers have discovered that polished adlay and adlay seed bran exhibit significant anticancer activity both in vitro and in vivo, including against lung cancer [14], colon cancer [15], and breast cancer [16]. In addition, adlay hulls are rich in bioactive small molecules, such as flavonoids, phenolic acids, and sterols [13,17,18]. Studies have demonstrated that adlay hulls exhibit significant inhibitory effects on human breast cancer cells (MCF-7) and cervical cancer cells (HeLa), with quercetin identified as one of the active components [17]. Additionally, research suggests that extracts from adlay hulls possess stronger antioxidant and antiproliferative activities compared to extracts from other parts of adlay and can induce apoptosis in U937 cells [19]. These findings underscore its potential as a therapeutic agent for AML. However, further research is required to identify its active compounds and elucidate the underlying mechanisms.

Due to the significant biological differences between AML and solid tumors, as well as the urgent need for novel therapeutic strategies for AML, this study investigates the mechanisms of action of adlay hull extract on AML cell lines U937 and HL-60. This research fills a gap in the study of adlay hull extract in hematologic malignancies and provides a novel candidate strategy for the development of natural therapeutics for AML. Furthermore, this study systematically analyzes the chemical composition of adlay hull extract using ultra-high-performance liquid chromatography–ion mobility spectrometry–quadrupole time-of-flight mass spectrometry (UPLC-IMS-QTOF-MS) and identifies its active components. This series of studies aims to evaluate the therapeutic potential of adlay hull extract in AML and to elucidate the mechanisms of action of its bioactive components, thereby laying a scientific foundation for its further investigation as a novel therapeutic candidate.

## 2. Materials and Methods

### 2.1. Plant Material

Adlay was collected from cultivated fields in Baoding, Hebei Province, China, during the harvest season of October 2023. The collection was conducted following standard procedures to ensure the seeds were mature and of high quality. The adlay hulls were obtained from mature seeds through mechanical separation and were purchased from Anguo Kangle Medicinal Herbs Store (Anguo, China) via the Taobao platform (https://e.tb.cn/h.6KLxyxykpLhUar2?tk=GJ25Vhsefri, accessed on 10 November 2023). The hulls were specifically purchased as already dehulled and dried.

### 2.2. Preparation of the AHE 

Adlay hulls (3.2 kg) were ground and passed through a 20-mesh sieve (aperture 0.94 mm). To prepare the ethanol extract, 95% ethanol was used as the solvent at a solid-to-liquid ratio of 1:5. The mixture was subjected to ultrasonic extraction at 25 °C with an ultrasonic power of 300 W for 2 h. The extract was then concentrated to dryness by rotary evaporation in a water bath at 20–25 °C. The residue was sequentially partitioned with n-hexane, ethyl acetate, and n-butanol from the aqueous solution. Each fraction was concentrated by rotary evaporation and dried under vacuum using a pump (IKA, Staufen, Germany). The fractions were concentrated to yield AHE-HEX (12.768 g, 0.40% *w*/*w*), AHE-EA (5.376 g, 0.17% *w*/*w*), and AHE-Bu (3.616 g, 0.11% *w*/*w*), which were stored at −20 °C.

### 2.3. Separation of AHE-EA

AHE-EA was separated by silica gel column chromatography (300–400 mesh). The column was manually packed using silica gel purchased from Qingdao Haiyang Chemical Co., Ltd. (Qingdao, China). The elution was performed using a gradient system with petroleum ether/ethyl acetate (Pe/EtOAc) for the initial phase, followed by dichloromethane/methanol (CHCl_2_/MeOH) for the later phase. (AHE-EA-A: 10% ethyl acetate in petroleum ether, *v*/*v*; AHE-EA-B: 25% ethyl acetate in petroleum ether, *v*/*v*; AHE-EA-C: 2–3% methanol in dichloromethane, *v*/*v*; AHE-EA-D: 10% methanol in dichloromethane, *v*/*v*; AHE-EA-E: 17% methanol in dichloromethane, *v*/*v*; AHE-EA-F: 50% methanol in dichloromethane transitioning to 100% methanol, *v*/*v*). Fractions with similar Rf values were pooled based on thin-layer chromatography (TLC) analysis, resulting in a total of six fractions (AHE-EA-A: 1.0361 g, 19.27%, *w*/*w*; AHE-EA-B: 1.1714 g, 21.78%, *w*/*w*; AHE-EA-C: 0.5416 g, 10.07%, *w*/*w*; AHE-EA-D: 1.1778 g, 21.90%, *w*/*w*; AHE-EA-E: 0.0532 g, 0.99%, *w*/*w*; AHE-EA-F: 1.148 g, 21.35%, *w*/*w*), which were then evaluated for their anti-AML activity, including effects on cell proliferation and apoptosis. Fractions with similar Rf values, as determined by thin-layer chromatography (TLC), were combined. A total of six fractions (AHE-EA-A to AHE-EA-F) were obtained and subsequently screened for anti-AML activity. (The procedure for preparing the AHE is shown in Figure 1).

### 2.4. UPLC-IMS-QTOF-MS Analysis of the AHE-EA-C Extract

The analysis was performed using an Acquity I-class UPLC system (Waters, Shanghai, China) coupled with a VION IMS-QTOF mass spectrometer (Waters, Shanghai, China). Chromatographic separations were carried out on a BEH C18 column (2.1 mm × 100 mm, 1.7 µm; Waters, Milford, MA, USA) equipped with a pre-column, maintained at 45 °C. The mobile phase consisted of 0.1% formic acid in water (A) and 0.1% formic acid in acetonitrile (B), with a flow rate of 0.4 mL/min. The gradient elution program was as follows: 0 min, 95% A and 5% B; 3 min, 80% A and 20% B; 10 min, 0% A and 100% B; 12 min, 0% A and 100% B; 15 min, 95% A and 5% B; and 20 min, 95% A and 5% B. The injection volume was 1 µL, and the needle wash solvent was H_2_O/ACN (10/90).

Mass spectrometric analysis was conducted in MSE mode with alternating low- and high-energy scans. Ionization was performed using electrospray ionization (ESI) in both positive and negative modes, with a capillary voltage of 2 kV and a cone voltage of 40 V. The desolvation gas flow rate was set at 900 L/h at 450 °C, and the cone gas flow rate was 50 L/h. The source temperature was maintained at 115 °C. Full-scan mass spectra were acquired in the range of *m*/*z* 50–1000 at a scan rate of 0.2 s. Collision energies were set at 6 eV for low-energy scans and 20–45 eV for high-energy scans. Real-time mass calibration was achieved using a lock mass of leucine enkephalin (250 pg/µL) infused at 10 µL/min, with data collected at intervals of 0.5 s and a sample time of 0.5 s.

### 2.5. Cell Culture and Cell Line

The U937, HL-60, HepG2, HT29, H1975, and 293T cell lines were obtained from the American Type Culture Collection (ATCC, Manassas, VA, USA). All cell lines were routinely tested to confirm that they were free of Mycoplasma. The U937, HL-60, and H1975 cell lines were cultured in RPMI-1640 (Gibco, Grand Island, NY, USA), whereas the HT29, HepG2, and 293T cell lines were cultured in DMEM (Gibco, Grand Island, NY, USA). All culture media were supplemented with 10% FBS with 100 U/mL penicillin and 100 μg/mL streptomycin, and the cells were cultured at 37 °C in 5% CO_2_. The cells were kept at low passages (3–5 passages) once obtained from vendors.

### 2.6. Antibodies and Reagents

Antibodies against the following proteins were used with the source and dilution ratios indicated: Cyclin D1 (HUABIO, Hangzhou, China, #ET1601-31, 1:5000); CDK4 (HUABIO, Hangzhou, China, #ET1612-23, 1:2000); PARP (HUABIO, Hangzhou, China, #ET1608-56, 1:2000); cleaved-PARP (HUABIO, Hangzhou, China, #ET1608-10, 1:500); Caspase 3 (CST, Danvers, MA, USA, #9662, 1:1000); cleaved caspase 3 (CST, Danvers, MA, USA, #9661, 1:1000); Bcl-2 (CST, Danvers, MA, USA, #4223, 1:1000); Bax (CST, Danvers, MA, USA, #5023, 1:1000); beat-actin (β-Actin) (HUABIO, Hangzhou, China, #EM21002, 1:10,000); Anti-rabbit lgG Fab2 (Sigma, St. Louis, MO, USA, #A0545, 1:10,000); and Antimouse lgG Fab2 (Sigma, St. Louis, MO, USA, #A4416, 1:10,000). Phosphate-buffered saline (PBS) washing buffer, Fetal bovine serum (FBS), Trypsin-EDTA solution, and Penicillin–Streptomycin solution (PS) (100×) were all purchased from Gibco (Grand Island, NY, USA). A Cell Counting Kit-8 (CCK-8), a cell cycle assay kit (Cat# C1052), an Annexin V-FITC apoptosis detection kit (Cat# C1062S), and RIPA lysis buffer were purchased from Beyotime (Shanghai, China). The Bicinchoninic acid (BCA) Protein assay kit was acquired from TIANGEN (Shanghai, China). The cocktail was obtained from Roche (Basel, Lewes, UK). Polyvinylidene fluoride (PVDF) membranes were purchased from Millipore (Billerica, MA, USA). All other commercially available reagents and solvents were sourced from Energy Chemical (Shanghai, China) and Shanghai Macklin Biochemical Technology Co., Ltd. (Shanghai, China), and were used without further purification.

### 2.7. Cell Viability Assay

The cells were cultured in 96-well plates at a density of 5 × 10^3^ cells/well. Cell viability was assessed with a CCK-8 Kit at the indicated time post-treatment according to the manufacturer’s instructions. To estimate the viability of the cells, the absorbance of 450 nm (OD450) was measured with a Microplate Reader (BIO-TEK, Inc., Winooski, VT, USA). The IC_50_ value was calculated by GraphPad Prism 8.0 software (San Diego, CA, USA).

### 2.8. Western Blot Analysis

Western blot analysis was performed as in our previous reports [20]. The cells were lysed in RIPA buffer with a complete protease inhibitor cocktail for 10 min on ice, followed by centrifugation at 4 °C. The concentration of proteins was determined by the BCA Protein Kit (TIANGEN, Shanghai, China). Equal amounts of total proteins were resuspended in loading buffer, boiled at 100 °C for 5 min, and separated by 10–15% sodium salt–polyacrylamide gel electrophoresis (SDS-PAGE). Proteins were transferred onto PVDF membranes (Merck Millipore, #IPFL00010, Darmstadt, Germany); then, the membranes were blocked with 5% non-fat dry milk, and the membranes were incubated with the indicated primary antibodies diluted in BSA buffer overnight at 4 °C. The membranes were washed three times in TBST, followed by 1 h incubation with secondary antibodies conjugated with horseradish peroxidase (HRP) at room temperature. Then, the membranes were washed before enhanced chemiluminescence. Immunoblots were visualized with the Bio-Rad ChemiDoc XRS system (Bio-Rad Laboratories, Hercules, CA, USA). Quantification was directly performed on the blot using Image Lab software (version 6.1).

### 2.9. Apoptosis and Cell Cycle Assay

Cell apoptosis was analyzed by an Annexin V-FITC Apoptosis Detection Kit (Beyotime, Shanghai, China). The cell cycle was analyzed by a PI Cell Cycle and Apoptosis Analysis Kit (Beyotime, Shanghai, China). The AML cells were seeded in 6-well plates at a density of 2 × 10^5^ cells per well and were treated with indicated different doses of AHE-EA-C for 48 h; DMSO served as vehicle control. After treatment, the cells were collected and stained with Annexin V-FITC and PI following the manufacturer’s protocol to analyze cell apoptosis and the cell cycle. Cell apoptosis and the cell cycle were detected by Beckman Coulter CytoFLEX flow cytometer (Brea, CA, USA). Apoptosis data were analyzed by FlowJo software (version 10.8.1), and cell cycle data were analyzed with Modfit LT 5.0 software.

### 2.10. EdU Incorporation Detection

Cell proliferation was assessed using a BeyoClick™ EdU-594 Cell Proliferation Kit (Beyotime, Cat# C0078S, Shanghai, China) according to the manufacturer’s instructions. Briefly, AML cells were seeded into 6-well plates at a density of 2 × 10^5^ cells per well and treated with different concentrations of AHE-EA-C for 48 h. Following treatment, the cells were incubated with 10 μM EdU solution for 2.5 h at 37 °C in a 5% CO_2_ atmosphere. After labeling, the cells were collected, fixed with 4% paraformaldehyde for 15 min, and permeabilized with 0.3% Triton X-100 for 10 min. Subsequently, the cells were stained with the Click reaction solution for 30 min at room temperature in the dark. Finally, the fluorescence intensity of the EdU-positive cells was detected using a Beckman Coulter CytoFLEX flow cytometer. The data were analyzed using FlowJo software.

### 2.11. Statistics Analysis

All the experimental data were analyzed by GraphPad Prism 8.0 (GraphPad Software Inc., San Diego, CA, USA). The results were presented as mean ± standard deviation (S.D.), and all the biological assays were conducted with *n* = 3 independent replicates. Statistical analysis was conducted using one-way ANOVA, with Tukey’s multiple-comparisons test applied. NS represents no significance, where * *p* < 0.05, ** *p* < 0.01, *** *p* < 0.001, and **** *p* < 0.0001.

## 3. Results

### 3.1. AHE-EA-C Inhibits the Cell Viability of AML Cells

To isolate the most effective anti-AML components from adlay hulls and evaluate their selective cytotoxicity, we systematically extracted the adlay hulls and assessed the inhibitory activity and cell proliferation effects of different extracts and their subfractions. Figure 1 illustrates the preparation process of the adlay hull ethanol extract. Using the same extraction protocol, three distinct fractions were obtained from adlay hulls (AHE-HEX, AHE-EA, and AHE-Bu), and three additional distinct fractions were obtained from polished adlay (PAE-HEX, PAE-EA, and PAE-Bu). The inhibitory effects and cell proliferation impacts of these six extracts were systematically evaluated in the H1975, HepG2, HT29, and HL-60 cell lines using the CCK-8 assay following a 72 h treatment period. The results showed that the AHE-EA fraction exhibited the most potent inhibitory effect on the AML cell line HL-60 and demonstrated a concentration-dependent suppression of cell proliferation (Appendix A). Therefore, the AHE-EA fraction was selected for further isolation. The AHE-EA fraction was subjected to silica gel column chromatography, yielding six subfractions (AHE-EA-A-AHE-EA-F). These subfractions were subsequently tested for their inhibitory activity and their effects on cell proliferation in AML cell lines (HL-60 and U937). The AHE-EA-C subfraction exhibited the most potent inhibitory effect (Figure 2A), with concentration-dependent suppression of cell viability, which was further corroborated by bright-field microscopy, revealing distinct morphological changes in the treated cells Specifically, with increasing concentrations of AHE-EA-C, both AML cell lines showed a marked reduction in cell number, along with membrane blebbing, cell shrinkage, cytoplasmic condensation, and irregular cell margins (Figure 2B). The IC_50_ values for the HL-60 and U937 cells were 29.86 ± 1.01 µM and 32.15 ± 1.79 µM, respectively (Figure 2C,D). Furthermore, the toxicity of AHE-E-A-C was evaluated on the human normal cell line 293T, with an IC_50_ value of 117.2 ± 3.33 μg/mL, which was significantly higher than that observed in the AML cell lines, indicating better biocompatibility and lower cytotoxicity. In conclusion, AHE-EA-C significantly inhibited the viability and proliferation of AML cells (U937 and HL-60).

### 3.2. AHE-EA-C Suppresses DNA Synthesis and Induces Cell Cycle Arrest in AML Cells

To further verify the antiproliferative activity of AHE-EA-C, an EdU incorporation assay was conducted to evaluate DNA synthesis in AML cells. Flow cytometry was used to quantify EdU incorporation levels after treatment with varying concentrations of AHE-EA-C. As shown in Appendix A, the right peak with higher fluorescence intensity represents actively proliferating cells undergoing DNA replication, whereas the left peak with lower fluorescence intensity corresponds to non-proliferating cells. In both the U937 and HL-60 cell lines, AHE-EA-C treatment led to a dose-dependent decrease in the proportion of EdU-positive cells, indicating that AHE-EA-C significantly suppresses DNA synthesis in AML cells.

To investigate whether the antiproliferative effect is mediated by AHE-EA-C-induced cell cycle arrest, flow cytometry was used to analyze the cell cycle distribution of AML cells, and Western blotting was performed to assess the expression levels of cell cycle regulatory proteins following AHE-EA-C treatment. As shown in Figure 3A,B, treatment with AHE-EA-C significantly increased the proportion of AML cells (HL-60 and U937) in the G0/G1 phase, accompanied by a subsequent reduction in the G2/M phase compared to the control. Additionally, Western blot analysis of cell cycle regulatory proteins, including CDK4 and Cyclin D1, was performed. As depicted in Figure 3C,D, quantitative assessment of CDK4 and Cyclin D1 expression revealed a dose-dependent decrease in AML cells (HL-60 and U937) relative to the control group. Collectively, these findings support the notion that G0/G1 phase arrest may play a crucial role in the AHE-EA-C-mediated antiproliferative effects in HL-60 and U937 cells.

### 3.3. AHE-EA-C Induces Cell Apoptosis in AML Cells

Apoptosis is widely recognized as a critical molecular mechanism in targeted cell death. To determine whether apoptosis plays a role in AHE-EA-C-induced cellular inhibition, we evaluated the apoptotic rate of AML cells following treatment with different concentrations of AHE-EA-C (0, 40, and 60 μg/mL) for 48 h. As illustrated in Figure 4A,B, exposure to AHE-EA-C resulted in a dose-dependent increase in late-stage apoptosis. Furthermore, the expression levels of apoptosis-related proteins, including PARP, Caspase 3, Bax, and Bcl-2, were assessed by immunoblotting. The results indicated that AHE-EA-C treatment dose-dependently upregulated the expression of cleaved caspase-3, cleaved-PARP, and Bax, while significantly downregulating Bcl-2 levels (Figure 4C,D).

### 3.4. UPLC-IMS-QTOF-MS Analysis of AHE-EA-C

In this study, mass spectrometry detection was performed using both positive ion mode and negative ion mode under the same liquid chromatography conditions. The base peak intensity chromatograms are shown in Figure 5A,B. Various data acquisition modes, including MSE and base peak intensity (BPI), were employed to obtain more precise mass spectral information. The MassLynx software (version 4.2 SCN 1049) system was used for data processing and analysis, which automatically matched the fragment ions. In this study, the in-house compound library developed by Shanghai Jiao Tong University, along with the Chinese Pharmacopoeia Database (CPD), PubChem, and SciFinder, were utilized to assist in the identification and matching of compounds. This method is suitable for comprehensive analysis and identification of complex samples and components, and it has widespread applications in modern mass spectrometry analysis. As a result, we significantly enhanced data analysis efficiency and reduced the complexity of compound identification. Ultimately, 52 compounds were identified in AHE-EA-C (Table 1).

### 3.5. PI3K/Akt Signaling Pathway Involvement in AHE-EA-C-Induced Cell Apoptosis

Although the PI3K/Akt pathway is widely activated across the entire AML population, including in leukemic cells at earlier developmental stages, constitutive activation of PI3K is detectable in 50% of acute myeloid leukemia samples. Therefore, we investigated the activation of the PI3K/Akt signaling pathway in AML cells following treatment with AHE-EA-C. AML cells were treated with AHE-EA-C for 48 h prior to analysis. As illustrated in Figure 6A,B, AHE-EA-C inhibited the phosphorylation of PI3K and Akt in a dose-dependent manner, with no significant change observed in the total levels of PI3K and Akt. Moreover, AHE-EA-C treatment caused a dose-dependent downregulation of Bcl-2 and upregulation of Bax, suggesting that its pro-apoptotic effects may be mediated via modulation of the PI3K/Akt signaling pathway.

## 4. Discussion

Inhibition of cell growth and promotion of apoptosis are crucial approaches in cancer therapy. In this study, we evaluated the inhibitory effects of AHE-EA on AML cells using the CCK-8 assay. The results showed that the AHE-EA-C fraction significantly inhibited the proliferation of U937 and HL-60 cells. Additionally, flow cytometry analysis revealed that after treatment with AHE-EA-C, the proportions of cells in the G1 and S phases significantly increased and decreased in a dose-dependent manner, respectively, suggesting that AHE-EA-C may inhibit cell proliferation by blocking the cell cycle. Further Annexin V-FITC apoptosis analysis showed that AHE-EA-C induced apoptosis in U937 and HL-60 cells, with the proportion of late apoptotic cells significantly increasing in a dose-dependent manner. Molecular level analysis further indicated that AHE-EA-C significantly decreased the expression of cell cycle markers (CDK4 and Cyclin D1) and anti-apoptotic proteins (Bcl-2), while the expression of pro-apoptotic proteins (cleaved caspase-3 and Bax) significantly increased. These results suggest that AHE-EA-C not only inhibits the proliferation of AML cells but also promotes apoptosis, indicating its potential as a strategy for the prevention and treatment of AML.

In AML patients, high expression of PI3K is closely associated with poor clinical outcomes and prognosis. PI3K regulates the downstream AKT/mTOR signaling pathway, influencing cell proliferation, survival, and metabolism, thereby promoting the progression of AML [10]. Specifically, PI3K activates AKT, which, in turn, regulates various proteins associated with cell cycle progression and apoptosis, such as Bcl-2 family proteins, Cyclin D, and p21Cip1. This process affects the survival and drug resistance of cancer cells [21]. Currently, the PI3K inhibitors Wortmannin and LY294002 have been widely used in preclinical models of AML, demonstrating potent cytotoxic effects in vitro. However, their insolubility in aqueous solutions and high toxicity have precluded the clinical application of these compounds [22,23,24,25]. Additionally, several natural products, such as terpenoids, flavonoids, and saponins, have been reported to exhibit PI3K inhibitory activity and can suppress the development of hepatocellular carcinoma. However, their research in AML has not been fully explored [26]. In this study, we found that AHE-EA-C significantly inhibits PI3K activity and exerts antiproliferative and pro-apoptotic effects by suppressing the PI3K/AKT signaling pathway. Furthermore, AHE-EA-C shows potential as a PI3K inhibitor in AML, warranting further investigation into its mechanism and clinical applicability.

Additionally, we performed chemical profiling of AHE-EA-C using UPLC-IMS-QTOF-MS and identified 52 compounds, primarily classified as fatty acids, flavonoids, steroids, and their derivatives. Notably, some components, such as isolappaol A, neohesperidin, picropodophyllotoxin, and cycloartanol, have been reported to exhibit significant anticancer activity, with isolappaol A and cycloartanol demonstrating specific inhibitory effects in leukemia models [27,28,29,30,31]. In recent years, multiple studies have highlighted the crucial role of flavonoids in AML treatment, acting through mechanisms such as inhibiting the VEGFR2 and PI3K/Akt signaling pathways, reducing mitochondrial membrane potential, downregulating Bcl-2 protein levels, and activating apoptosis-related proteins, including caspase-9 and caspase-3 [32,33,34,35]. Given that flavonoids are the major constituents of AHE-EA-C, these findings suggest that flavonoids may serve as the key bioactive components responsible for its anti-AML activity. Overall, this study reveals the abundant presence of flavonoids and other bioactive compounds in AHE-EA-C and supports its role as an important source of anti-AML agents derived from adlay hulls, providing a scientific basis for further investigations into its underlying mechanisms.

This study demonstrates that adlay hull extract significantly inhibits AML cell proliferation and promotes apoptosis by modulating the PI3K/Akt signaling pathway (Figure 7). Additionally, key bioactive compounds with anti-AML activity were identified in the extract. Future research will focus on the isolation and individual evaluation of these identified compounds to confirm their anticancer activity and elucidate their molecular mechanisms of action. Furthermore, additional studies will be needed to assess the efficacy of adlay hull extract and its active components in NOD/SCID mice engrafted with human AML cells. Meanwhile, pharmacokinetic and clinical studies of key compounds will be conducted to evaluate their absorption, distribution, metabolism, and excretion (ADME) properties and explore their therapeutic potential and safety in AML treatment.

## 5. Conclusions

In conclusion, this study demonstrates that the AHE-EA-C extract exerts significant antiproliferative and pro-apoptotic effects on AML cells (U937 and HL-60) by inhibiting the PI3K/Akt signaling pathway. UPLC-IMS-QTOF-MS analysis identified 52 bioactive compounds in AHE-EA-C, among which isolappaol A, neohesperidin, picropodophyllotoxin, and cycloartanol have been reported to possess potential anticancer and anti-AML activities. The findings indicate that flavonoids are among the major constituents of AHE-EA-C and may regulate AML cell proliferation and apoptosis through modulation of the PI3K/Akt pathway. Based on these results, the AHE-EA-C extract and its key bioactive components hold promising potential for AML treatment and may serve as functional foods or pharmaceutical supplements, offering a natural and low-toxicity therapeutic option for AML patients. Future studies will further investigate the efficacy and specific mechanisms of AHE-EA-C in in vivo models to facilitate its clinical application.

## Figures and Tables

**Figure 1 cimb-47-00358-f001:**
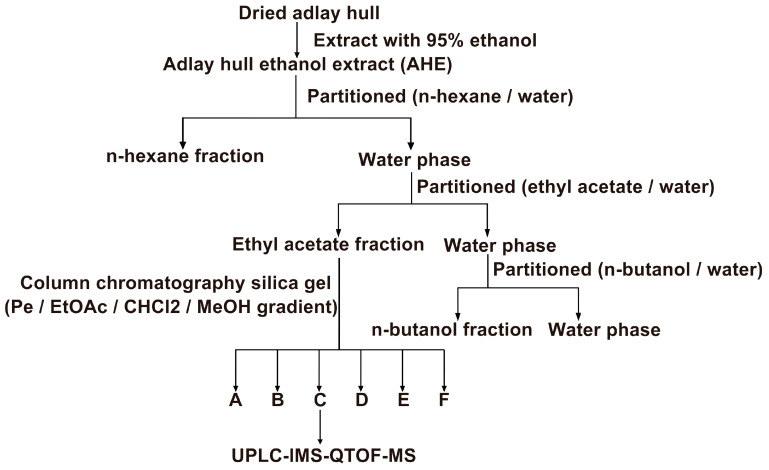
Scheme for the preparation of anti-AML fractions and compounds from adlay hulls.

**Figure 2 cimb-47-00358-f002:**
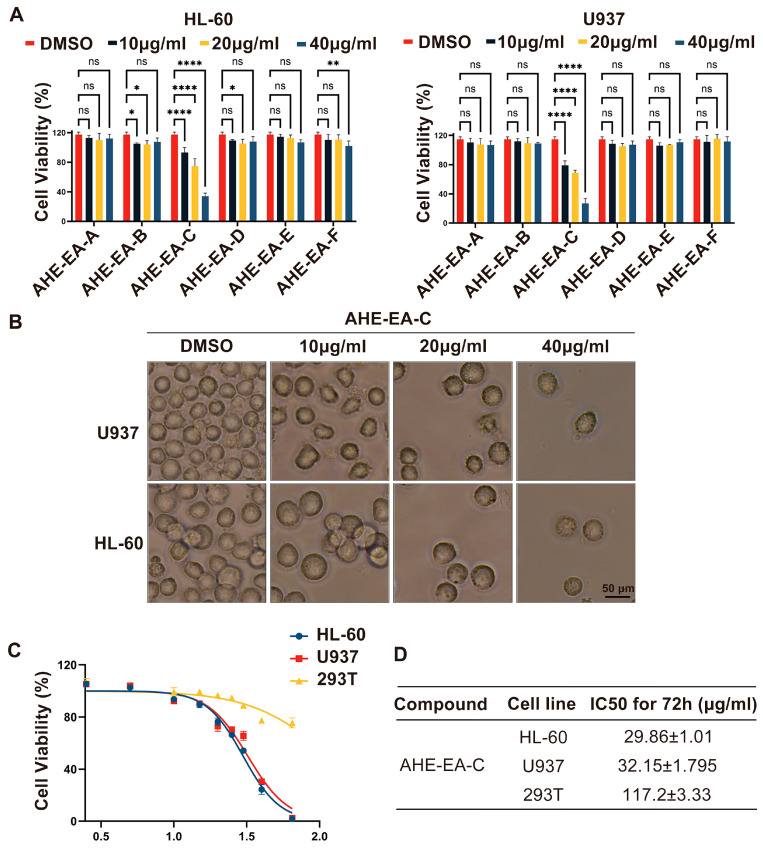
The impact of AHE-EA-C on viability and proliferation inhibition in AML cells. (**A**) The cell viability of HL-60 and U937 cells treated with different subfractions of AHE-EA (AHE-EA-A-AHE-EA-F), at concentrations of 10 μg/mL, 20 μg/mL, and 40 μg/mL, was assessed using the CCK-8 assay after a 72 h treatment. DMSO was used as the control. (**B**) Cell morphology of U937 and HL-60 cells was observed after treatment with DMSO or AHE-EA-C (10, 20, and 40 μg/mL) for 72 h. The cells were examined and photographed using an inverted microscope under bright field illumination (magnification: 400×, scale bar: 50 μm). (**C**) U937, HL-60, and 293T were treated with the indicated concentrations of AHE-EA-C (2.5, 5, 10, 15, 20, 25, 30, 40, and 65 μg/mL) or DMSO for 72 h, and cell viability was assessed using the CCK-8 assay and are shown as relative viability compared to the untreated control. Each test was performed in triplicate. (**D**) The IC_50_ values of AHE-EA-C in HL-60, U937, and 293T cells are presented as the mean ± SD. All data are shown as mean ± SD, *n* = 3, two-way ANOVA, * *p* < 0.05, ** *p* < 0.01, **** *p* < 0.0001, and ns = not significant.

**Figure 3 cimb-47-00358-f003:**
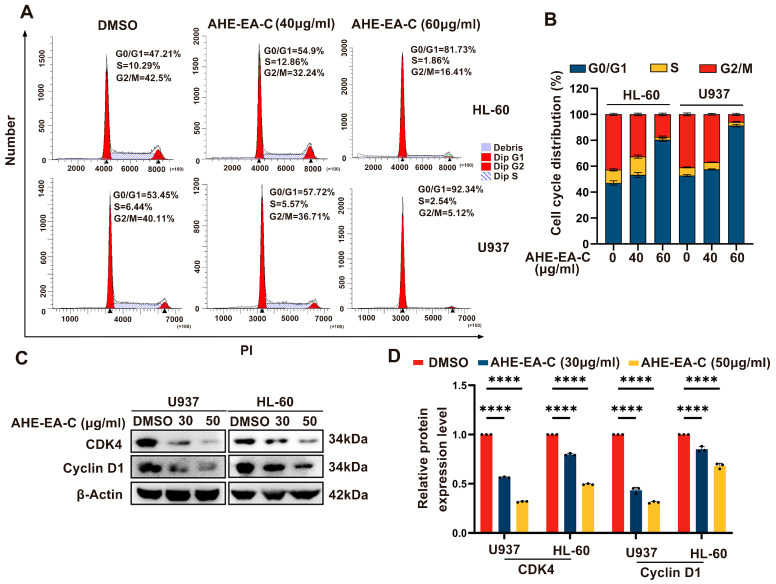
Cell cycle inhibition by AHE-EA-C in AML cells. (**A**) Flow cytometry revealed the cell cycle distribution of HL-60 and U937 cells treated with the indicated concentrations of AHE-EA-C (0, 40, and 60 μg/mL) for 48 h. (**B**) The statistical analysis of cell cycle distribution. (**C**) Immunoblot analysis was performed to examine the cell cycle distribution of U937 and HL-60 cells following 48 h incubation with the indicated concentrations of AHE-EA-C (0, 40, and 60 μg/mL). Protein levels of CDK4 and Cyclin D1 were quantified by Western blotting, with β-actin serving as the internal loading control. (**D**) Quantitative analysis of CDK4 and Cyclin D1 expression following 48 h treatment with AHE-EA-C at concentrations of 0, 40, and 60 μg/mL. All data are shown as mean ± SD, *n* = 3, two-way ANOVA, **** *p* < 0.0001.

**Figure 4 cimb-47-00358-f004:**
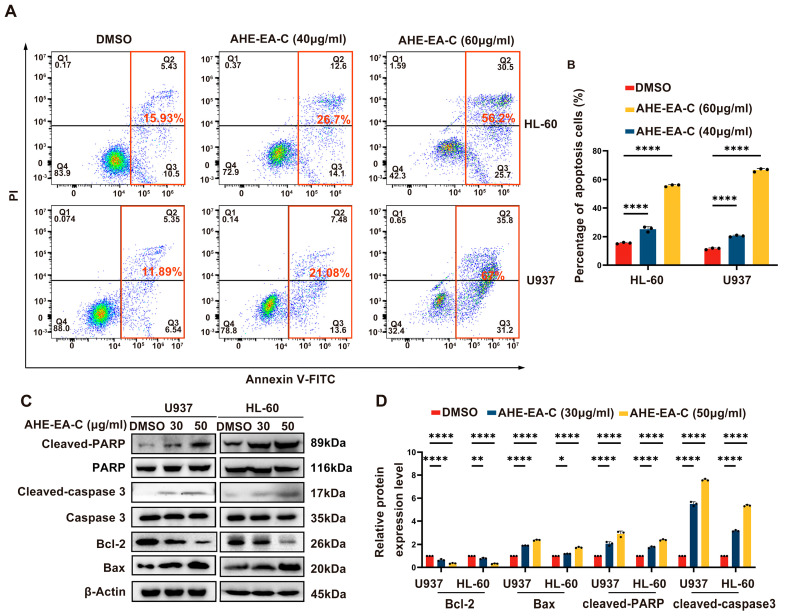
Induction of apoptosis in AML cells by AHE−EA−C treatment. (**A**) Apoptosis in HL−60 and U937 cells treated with the specified concentrations of AHE−EA−C (0, 40, and 60 μg/mL) for 48 h was assessed using the Annexin−V FITC/PI double-staining assay. (**B**) Quantitative analysis of cellular apoptosis following 48 h treatment with AHE−EA−C at concentrations of 0, 40, and 60 μg/mL. (**C**) Immunoblot analysis was performed to assess the expression levels of PARP and other apoptosis−related biomarkers in U937 and HL−60 cells following 48 h incubation with the indicated concentrations of AHE−EA−C (0, 40, and 60 μg/mL). Protein levels of PARP, cleaved−PARP, caspase 3, cleaved−caspase 3, Bcl-2, and Bax were measured by Western blotting, with β−actin used as the internal loading control. (**D**) Quantitative analysis of PARP, cleaved−PARP, caspase 3, cleaved caspase 3, Bcl−2, and Bax expression following 48 h treatment with AHE−EA−C at concentrations of 0, 40, and 60 μg/mL. All data are shown as mean ± SD, *n* = 3, two-way ANOVA, * *p* < 0.05, ** *p* < 0.01, **** *p* < 0.0001.

**Figure 5 cimb-47-00358-f005:**
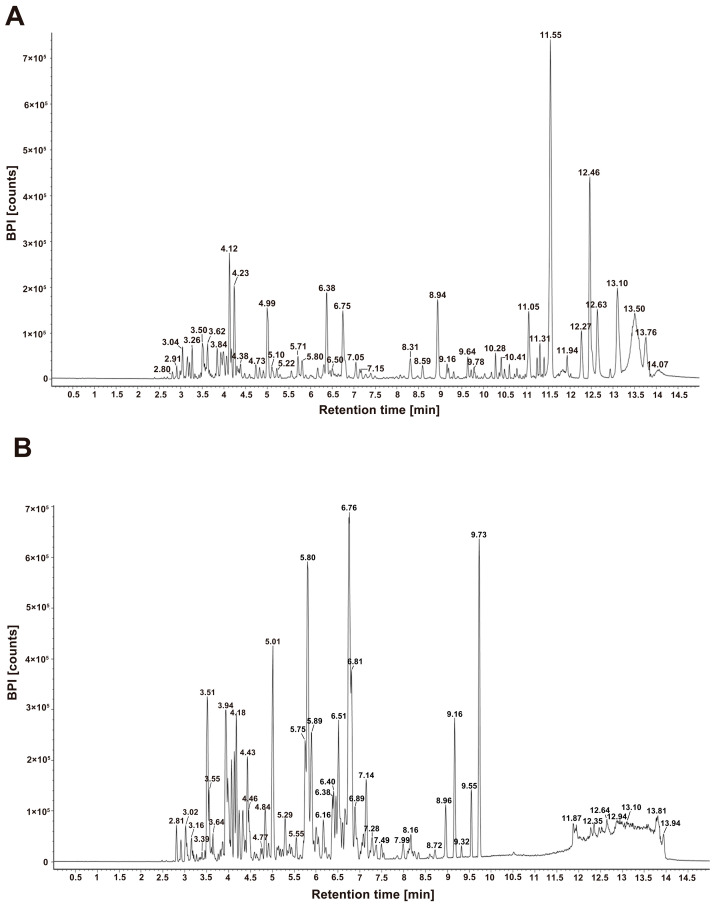
The base peak intensity chromatogram (BPI) of AHE-EA-C in positive (**A**) and negative (**B**) ion mode.

**Figure 6 cimb-47-00358-f006:**
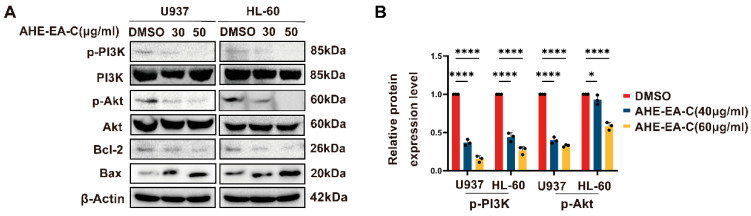
AHE-EA-C induces apoptosis in AML cells via the PI3K/Akt pathway. (**A**) Immunoblotting was performed to assess the phosphorylation status of the PI3K/Akt pathway in U937 and HL-60 cells following 48 h of incubation with the indicated concentrations of AHE-EA-C (0, 30, and 50 μg/mL). The protein levels of p-PI3K, PI3K, p-Akt, Akt, Bcl-2, and Bax were analyzed by Western blotting, with β-actin used as the endogenous loading control. (**B**) The band densities of p-PI3K, PI3K, p-Akt, and Akt were quantified following treatment with AHE-EA-C at concentrations of 0, 30, and 50 μg/mL for 48 h. All data are shown as mean ± SD, n = 3, two-way ANOVA, * *p* < 0.05, **** *p* < 0.0001.

**Figure 7 cimb-47-00358-f007:**
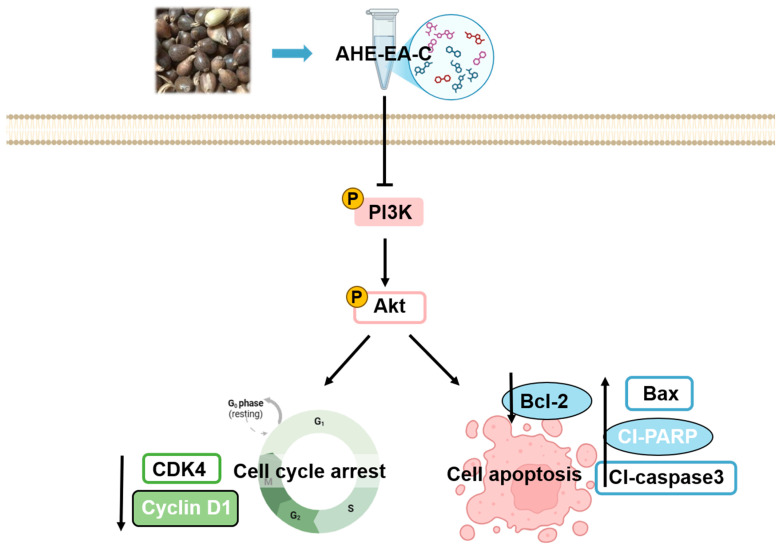
Schematic diagram illustrating the proposed mechanism by which AHE-EA-C exerts its anti-leukemic effects in AML cells. The T-shaped arrows represent inhibition, while the pointed arrows indicate activation or stimulation. Green-labeled proteins (CDK4, Cyclin D1) are associated with cell cycle regulation, and blue-labeled proteins (Bcl-2, Cl-PARP, Cl-caspase 3) are associated with apoptosis. AHE-EA-C treatment inhibits the PI3K/Akt signaling pathway, leading to cell cycle arrest and apoptosis. Specifically, AHE-EA-C downregulates the phosphorylation of PI3K and Akt, reducing the expression of cell cycle regulators CDK4 and Cyclin D1, which results in G1 phase arrest. Additionally, AHE-EA-C decreases the expression of the anti-apoptotic protein Bcl-2 while increasing the expression of the pro-apoptotic protein Bax. This shift in the apoptotic balance leads to the activation of cleaved caspase-3 (Cl-Caspase-3) and the cleavage of PARP (Cl-PARP), ultimately inducing apoptosis.

**Table 1 cimb-47-00358-t001:** Identification of chemical constituents of AHE-EA-C by UPLC-IMS-QTOF-MS. (The detailed information can be found from Appendix A).

No	Component Name	Formula	Neutral Mass (Da)	Observed Neutral Mass (Da)	Observed (*m*/*z*)	Mass Error (ppm)	Adducts	Observed RT (min)	Fragments (*m*/*z*)
1	4-Methyl ester octanoic acid	C_9_H_16_O_2_	156.11503	156.1146	201.1128	−2.3	+HCOO	4.46	201.1127, 223.0944, 202.1162, 183.1023
2	14-Methyl hexadecanoic acid	C_17_H_34_O_2_	270.2559	270.2555	315.2537	−1.3	+HCOO	9.55	315.2537, 485.3475, 433.2594, 313.2381, 121.0291
3	2″-O-Rhamno sylicariside Ⅱ	C_33_H_40_O_14_	660.2418	660.2418	659.2346	0	H	4.37	151.0035, 207.0658, 255.0659, 287.0557, 343.2124, 433.1505, 567.1871
4	3-Acetyl-3,4-dihydro5,6-dimethoxy-2(1)H-benzopyrone	C_13_H_14_O_5_	250.0841	250.084	295.0822	−0.5	+HCOO	3.61	151.0396, 165.0552, 195.0657, 295.0822, 327.1237, 377.1241, 389.1243, 520.0681
5	3-Hydroxy-5,7,8,3′,4′-pentamethoxy flavone	C_20_H_20_O_8_	388.1158	388.1159	387.1086	0.1	H	3.87	116.0503, 162.0318, 177.0553, 195.0656, 265.0713, 359.1131, 388.1116, 417.1190
6	5,7,4′-Trihydroxy flavanone	C_15_H_12_O_5_	272.0685	272.0682	317.0664	−0.7	+HCOO	4.03	145.0867, 149.0968, 237.1126, 289.0715, 318.0697, 317.0665, 555.2234
7	5,8,4′-Trihydroxy-6,7-dimethoxyflavone	C_17_H_14_O_7_	330.074	330.0737	329.0665	−0.6	H	5.11	329.06646, 327.21731, 343.21248, 351.13495
8	Citflavanone	C_20_H_18_O_5_	338.1154	338.1153	337.108	−0.5	−H	4.74	337.10799, 325.10786, 355.11863, 201.11282, 279.06556, 195.06556, 307.06081
9	Dibutyl sebacate	C_18_H_34_O_4_	314.2457	314.2453	313.2381	−1.2	−H	8.96	313.23806, 293.21119, 311.22243, 187.09721, 449.25460, 558.11949, 183.13846, 116.92812, 295.22724, 367.15737, 496.11874
10	Dihydrokaempferol	C_15_H_12_O_6_	288.0634	288.0631	287.0558	−0.9	−H	4.38	287.05567, 135.04469, 151.00325, 285.04011, 343.21240
11	Forsythoside C	C_29_H_36_O_16_	640.20034	640.1999	685.1981	−0.6	+HCOO	4.35	119.04979, 150.03161, 116.92809, 160.01604, 535.19679
12	Isolappaol A	C_30_H_32_O_9_	536.20463	536.2046	535.1974	0	−H	4.84	150.03154, 180.04203, 116.92802, 209.08061, 267.15932, 355.11743, 503.17017
13	Medioresinol	C_21_H_24_O_7_	388.1522	388.152	433.1502	−0.6	+HCOO	4.07	179.03404, 150.03152, 165.05503, 167.02069, 192.04202, 431.13455, 433.15004
14	Methyl 7,10-hexadecadienoate	C_17_H_30_O_2_	266.22458	266.224	311.2222	−1.7	+HCOO	7.37	145.02892, 215.12836, 293.21171, 309.20681, 327.21745
15	Moupinamide	C_18_H_19_NO_4_	313.13141	313.1309	312.1236	−1.6	−H	4.24	155.10742, 162.03155, 135.04453, 147.04450, 190.05034, 212.12878, 329.13872
16	Nonanedioic acid	C_9_H_16_O_4_	188.10486	188.1043	187.097	−3.2	−H	3.94	125.09676, 187.09685, 209.07896, 209.07928, 403.13973
17	N-trans-Coumaroyltyramine	C_17_H_17_NO_3_	283.12084	283.1206	282.1133	−0.9	−H	4.13	119.04979, 166.02619, 241.10766, 283.11630, 282.11300
18	Picropodophyllotoxin	C_22_H_22_O_8_	414.13147	414.1313	413.124	−0.4	−H	5.39	116.92804, 209.11784, 193.04979, 134.03676, 327.21701
19	Sesamol	C_7_H_6_O_3_	138.03169	138.0315	137.0242	−1.5	−H	3.02	137.02374, 160.04008, 192.95847, 327.96715, 328.97483
20	Sparassol	C_10_H_12_O_4_	196.07356	196.073	241.0712	−2.3	+HCOO	3.61	136.01618, 149.02388, 150.03157, 165.03961, 165.05510
21	Suberic acid	C_8_H_14_O_4_	174.08921	174.0889	173.0816	−1.9	−H	3.55	111.08115, 173.08116, 195.06525, 165.01885, 229.05734
22	Syringaldehyde	C_9_H_10_O_4_	182.05791	182.0575	181.0502	−2.5	−H	2.81	120.02130, 148.01618, 163.03969, 181.05018, 212.06074
23	Thujaplicatin methyl ether	C_21_H_24_O_8_	404.14712	404.147	403.1397	−0.3	−H, +HCOO	3.96	99.92543, 178.02638, 209.07896, 243.03129, 428.11357
24	Tianshic acid	C_18_H_34_O_5_	330.24062	330.2402	329.2329	−1.4	−H	6.51	171.10215, 137.09681, 199.1331, 311.22228, 330.23628
25	Xanthoxylin	C_10_H_12_O_4_	196.07356	196.0732	241.0714	−1.5	+HCOO	2.91	137.02421, 193.05011, 211.06081, 242.07484, 163.00314
26	11-Eicosenonic acid	C_20_H_38_O_2_	310.2872	310.2855	333.2747	−5	+Na	8.31	79.06537, 137.06517, 199.16110, 215.15830, 259.19084
27	1-Hydroxy-2,3,4,5-tetramethoxyxanthone	C_17_H_16_O_7_	332.0896	332.0907	355.0799	3	+Na	12.64	147.06935, 281.05857, 355.07989, 267.00625, 429.10015
28	Benzeneuropyl acetate	C_11_H_14_O_2_	178.0994	178.1001	179.1074	4.2	+H	2.98	128.96958, 179.10727, 187.07820, 161.09407, 133.09435
29	Bis(2-ethylhexyl) phthalate	C_24_H_38_O_4_	390.277	390.2766	391.2839	−1	+H	9.80	185.13265, 229.16589, 277.24348, 351.28753, 495.37838
30	Cyclomargenol	C_32_H_54_O	454.4175	454.4155	472.4493	−4.3	0	9.7	91.06347, 261.24856, 263.26444, 339.28559, 498.40146
31	Spinasterone	C_29_H_46_O	410.3549	410.3575	433.3467	6.1	+Na	8.30	137.06517, 199.16110, 215.15824
32	Δ5-Pregnene-3β,17α,20α-diol	C_21_H_34_O_2_	318.25588	318.2561	341.2453	0.6	+Na	7.39	185.14434, 201.14221, 245.17512, 341.24465
33	12-Acetoxyl-9-octadecenoate oleic acid methyl ester	C_21_H_40_O_4_	356.29266	356.2945	374.3283	4.8	0	11.15	235.06734, 327.27289, 374.32829, 459.33356, 504.39409
34	13,17-Epoxy alisol A	C_30_H_50_O_6_	506.36074	506.3609	507.3681	0.3	+H	10.79	121.07382, 223.08053, 368.45337, 502.41217, 582.51805
35	2,7-Dihydroxy-1-(p-hydroxybenzyl)-4-methoxy-9,10-dihydrophenanthrene	C_22_H_20_O_4_	348.13616	348.1381	371.1273	5.1	+Na	11.05	151.03450, 223.07906, 371.12727
36	Aloenin	C_19_H_22_O_10_	410.1213	410.1222	411.1295	2.2	+H	10.41	191.01721, 223.08286, 281.07520, 411.12947
37	Aurantiamide acetate	C_27_H_28_N_2_O_4_	444.20491	444.2024	445.2097	−5.6	+H	3.52	147.06450, 177.07956, 291.12511, 445.20969
38	Cycloartanol	C_30_H_52_O	428.40182	428.4048	451.394	6.6	+Na	11.07	89.06575, 297.26078, 341.28964, 429.34777
39	Deoxycholic acid	C_24_H_40_O_4_	392.29266	392.2951	393.3023	6.1	+H	9.71	147.05908, 263.26444, 497.39736
40	Dihydrosterculic acid	C_19_H_36_O_2_	296.27153	296.2706	297.2778	−3.3	+H	9.16	279.26539, 337.27549, 229.17100
41	Grosvenorine	C_33_H_40_O_19_	740.21638	740.2139	758.2477	−3.3	+NH_4_, +H	12.54	103.95861, 262.18817, 283.05735
42	Hydroxyobtustyrene	C_16_H_16_O_3_	256.10994	256.1112	257.1185	4.8	+H	4.34	177.08022, 257.11846, 323.19160, 387.20039
43	Isopropyl salicylate	C_10_H_12_O_3_	180.07864	180.0795	181.0868	4.7	+H	3.04	163.07368, 181.08677, 203.07197, 145.06040
44	Lavandulifolioside	C_34_H_44_O_19_	756.24768	756.2497	774.2836	2.7	0	12.36	355.08246, 273.17689, 222.09291
45	Methyl succinate	C_6_H_10_O_4_	146.05791	146.0578	147.065	−1	+H	4.13	147.06504, 219.13325, 284.16923, 353.19863
46	Neohesperidin	C_28_H_34_O_15_	610.18977	610.1915	628.2254	2.8	+NH_4_, +H, +Na	11.94	207.04191, 355.08661, 445.14091
47	Periplocoside N	C_27_H_44_O_6_	464.31379	464.3147	487.3039	1.8	+Na	6.44	91.06782, 147.06581, 193.15130, 275.24208, 293.25526, 487.30388
48	Pinoresinol dimethyl ether II	C_22_H_26_O_6_	386.17294	386.1738	404.2076	2.2	0	3.95	91.06711, 161.08254, 177.07994, 233.11387, 429.21312
49	Pyrophaeophorbide A	C_33_H_34_N_4_O_3_	534.26309	534.2606	557.2498	−4.5	+Na	6.37	177.08127, 223.09717, 557.24981
50	Scopolin	C_16_H_18_O_9_	354.09508	354.0935	355.1008	−4.4	+H	10.41	191.01721, 207.05011, 223.08286, 281.07540
51	Terrestribisamide	C_24_H_28_N_2_O_6_	440.19474	440.1934	458.2273	−2.8	0	4.45	173.15773, 177.08053, 159.14007, 229.15594, 247.16903, 305.18080
52	Tribulusamide B	C_36_H_34_N_2_O_9_	638.22643	638.2249	656.2587	−2.3	0	11.94	147.07213, 207.04155, 267.01134, 281.06398

## Data Availability

The original contributions presented in this study are included in the article/Appendix A. Further inquiries can be directed to the corresponding authors.

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
