# Peer review of "Ethanol Extract of Adlay Hulls Suppresses Acute Myeloid Leukemia Cell Proliferation via PI3K/Akt Pathway Inhibition"

_cimb, 2025, doi:10.3390/cimb47050358_

Round 1

Reviewer 1 Report

Comments and Suggestions for Authors

The manuscript is interesting and clear. However, I have the following comments.

The title should focus on the AML cell line (HL60) in which the compounds were evaluated, since this type of leukemia is highly heterogeneous and does not include other subtypes of the pathology.

It is important to review the characteristics of the cell lines used, especially U937 that is not AML and in ATCC it is referred to a Histiocytic Lymphoma, this point also indicated in reference 21 (Human Histolytic Lymphoma Monocytic Cells).  Additionally the 293T cell line, which in the text appears to correspond to normal hematopoietic cells is a human embryonic kidney cell line.

In the results it is necessary to describe the morphological changes associated with compounds treatment, since they are not very clear.

I suggest you specify why a different concentration than the IC50 was used for all experiments (including those in the supplementary figures) It is desirable to discuss why the compound AHE-EA-C seems to have a lower effect (even at concentrations of 100ug) on ​​H1975, HepG2 and HT29, when it has been suggested that this cells use the PI3K/AKT pathway for their proliferation

Author Response

Major concerns:

  1. The title should focus on the AML cell line (HL60) in which the compounds were evaluated, since this type of leukemia is highly heterogeneous and does not include other subtypes of the pathology. It is important to review the characteristics of the cell lines used, especially U937 that is not AML and in ATCC it is referred to a Histiocytic Lymphoma, this point also indicated in reference 21 (Human Histolytic Lymphoma Monocytic Cells).

Response:

Thank you for your attention to our work. We understand that the U937 cell line was initially classified as a lymphoid tissue origin in the ATCC, but numerous studies have demonstrated that this cell line exhibits typical myeloid characteristics, capable of differentiating into monocytes and macrophages, thus making it widely used in research related to AML. Particularly in exploring the mechanisms of AML cell apoptosis, chemotherapy sensitivity, metabolic regulation, and targeted therapy, U937 has proven to be a highly applicable experimental model. For instance, Chiou et al. found that BCL2L1 inhibitors induce apoptosis in U937 cells by suppressing MCL1 transcription [1]; Jia et al. confirmed that hsa-miR-12462 can enhance the sensitivity of U937 to cytarabine by targeting SLC9A1 [2]; Zhang et al. used this cell line to reveal that metformin can induce cell cycle arrest via the AMPK/mTOR pathway[3]; Yakymiv et al. indicated that CD157 regulates Mcl-1 expression, promoting the survival of U937 and enhancing its resistance to chemotherapy drugs [4]. These studies support the broad applicability of U937 in AML mechanism research. Therefore, we believe that the use of U937 as an AML cell model in this study is well-supported by literature and is rational.

  1. Additionally the 293T cell line, which in the text appears to correspond to normal hematopoietic cells is a human embryonic kidney cell line.

Response:

We sincerely acknowledge that 293T cells are derived from human embryonic kidney cells and are not hematopoietic in origin. However, due to ethical and technical limitations, the isolation and maintenance of normal human hematopoietic stem or progenitor cells in vitro pose considerable challenges. 293T cells are commonly used as controls in drug screening due to their ease of use, stability, and non-malignant background.

Previous studies have reported the use of 293T cells as controls in AML research. For example, in a 2023 study, Liu et al. employed 293T cells as a non-cancerous control to evaluate the cytotoxic effects of Z-ligustilide on AML HL-60 cells [5]. Their results demonstrated that Z-ligustilide exhibited selective toxicity toward HL-60 cells while exerting minimal effects on 293T cells, supporting the rationale for using 293T cells as a reference in preliminary drug screening.

Although 293T cells are not derived from the hematopoietic system, their application as surrogate normal cells for assessing off-target toxicity in AML studies is supported by existing literature.

  1. In the results it is necessary to describe the morphological changes associated with compounds treatment, since they are not very clear.

Response:

Thank you very much for your valuable comments regarding the morphological description of the cells. To enhance clarity and facilitate reference, we have added a detailed description of the morphological observations in the revised manuscript (lines 248–251), which has been highlighted in red. The specific content is as follows:

"Specifically, with increasing concentrations of AHE-EA-C, both AML cell lines showed a marked reduction in cell number, along with membrane blebbing, cell shrinkage, cytoplasmic condensation, and irregular cell margins (Figure 2B)."

We believe that this additional information will help readers better understand the morphological effects of AHE-EA-C treatment on AML cells. Once again, we sincerely appreciate your insightful suggestions and continued support.

  1. I suggest you specify why a different concentration than the IC50 was used for all experiments (including those in the supplementary figures).

Response:

Thank you for your valuable suggestions. In the experimental design, we selected concentrations slightly above the ICâ‚…â‚€ values for the following reasons: lower concentrations (1×ICâ‚…â‚€) typically require longer drug exposure to produce significant biological effects, whereas higher concentrations (2–3×ICâ‚…â‚€) can enhance signal clarity within shorter treatment durations. Furthermore, considering the maximum solubility of AHE-EA-C in DMSO is approximately 130 mg/mL, this limits the highest dose that can be practically applied. Consequently, for flow cytometry analysis, including cell cycle and apoptosis experiments, we selected concentrations of 40 and 60 μg/mL to ensure sufficient cellular responses could be observed within 48 hours.

For Western blot experiments, due to the rapid response of protein expression, we employed lower concentration gradients (30 and 50 μg/mL). Notably, this design also considers the ICâ‚…â‚€ values of both AML cell lines—32.15 ±â€¯1.80 μg/mL for U937 and 29.86 ±â€¯1.01 μg/mL for HL-60—and uses 30 μg/mL as a representative low-dose close to the ICâ‚…â‚€ for both. The rationale for selecting concentrations of 25 and 100 μg/mL in the supplementary experiments is that the ICâ‚…â‚€ value of the tested compound had not yet been determined; thus, multiple concentration points were used for preliminary activity screening. Overall, this concentration selection strategy was intended to balance solubility constraints with experimental needs, thereby ensuring the reliability and interpretability of the results.

  1. It is desirable to discuss why the compound AHE-EA-C seems to have a lower effect (even at concentrations of 100ug) on H1975, HepG2 and HT29, when it has been suggested that this cells use the PI3K/AKT pathway for their proliferation.

Response:

Thank you for your insightful comments. Although the PI3K/AKT pathway is indeed associated with the proliferation of various cancer types, including H1975 (non-small cell lung cancer), HepG2 (hepatocellular carcinoma), and HT29 (colorectal cancer), the differential sensitivity of these solid tumor cell lines to AHE-EA-C compared to AML cells may be attributed to several factors:

  1. Resistance mechanisms in solid tumors: Solid tumors frequently develop resistance to PI3K/AKT pathway inhibitors through various mechanisms. For example, in hepatocellular carcinoma, long-term exposure to sorafenib can induce resistance that is characterized by PI3K/AKT pathway activation and epithelial-mesenchymal transition (EMT), and inhibition of this pathway with LY294002 can partially reverse this resistance phenotype [6]. Similarly, in non-small cell lung cancer, overexpression of HSPA12B promotes resistance to cisplatin by activating the PI3K/AKT/NF-κB signaling cascade, leading to increased cell survival and reduced apoptosis [7]. These resistance mechanisms may explain why PI3K/AKT-targeted treatments exhibit limited efficacy in certain solid tumors compared to hematological malignancies.
  2. Pharmacokinetic and microenvironmental factors: The tumor microenvironment of solid tumors can influence drug efficacy through factors such as hypoxia, extracellular matrix composition, and interstitial pressure, all of which can impede drug penetration and distribution. For example, in small cell lung cancer, activation of the PI3K/AKT/mTOR pathway has been shown to be closely associated with resistance to therapy, in part due to its interaction with hypoxia and altered glucose metabolism mediated by HIF-1 signaling [8]. This metabolic reprogramming contributes to therapeutic resistance by affecting bioenergetic processes and enhancing cellular survival in low-oxygen conditions. In addition, differences in drug uptake, intracellular metabolism, and efflux mechanisms between cell types may further alter the intracellular concentration of AHE-EA-C, potentially reducing its efficacy in solid tumor cells compared to hematological malignancies.

Collectively, the reduced efficacy of AHE-EA-C in solid tumors may stem from their intrinsic resistance mechanisms and microenvironmental barriers, which collectively diminish therapeutic sensitivity compared to AML cells.

References:

[1]  J.-T. Chiou, Y.-Y. Wu, Y.-C. Lee, L.-S. Chang, BCL2L1 inhibitor A-1331852 inhibits MCL1 transcription and triggers apoptosis in acute myeloid leukemia cells, Biochemical Pharmacology 215 (2023) 115738. https://doi.org/10.1016/j.bcp.2023.115738.

[2]  Y. Jia, W. Liu, H.-E. Zhan, X.-P. Yi, H. Liang, Q.-L. Zheng, X.-Y. Jiang, H.-Y. Zhou, L. Zhao, X.-L. Zhao, H. Zeng, Roles of hsa-miR-12462 and SLC9A1 in acute myeloid leukemia, J Hematol Oncol 13 (2020) 101. https://doi.org/10.1186/s13045-020-00935-w.

[3]  Y. Zhang, J. Li, W. Shi, Metformin Inhibits Acute Myeloid Leukemia Cells Growth through the AMPK/mTOR Pathway and Autophagic Regulation, Blood 140 (2022) 6175. https://doi.org/10.1182/blood-2022-167024.

[4]  Y. Yakymiv, S. Augeri, C. Bracci, S. Marchisio, S. Aydin, S. D’Ardia, M. Massaia, E. Ferrero, E. Ortolan, A. Funaro, CD157 signaling promotes survival of acute myeloid leukemia cells and modulates sensitivity to cytarabine through regulation of anti-apoptotic Mcl-1, Sci Rep 11 (2021) 21230. https://doi.org/10.1038/s41598-021-00733-5.

[5]  G. Liu, Z. Chen, L. Yang, Y. Rong, Q. Wang, L. Li, Q. Lu, M. Jiang, H. Qi, Z-ligustilide preferentially caused mitochondrial dysfunction in AML HL-60 cells by activating nuclear receptors NUR77 and NOR1, Chin Med 18 (2023) 123. https://doi.org/10.1186/s13020-023-00808-7.

[6]  H. Zhang, Q. Wang, J. Liu, H. Cao, Inhibition of the PI3K/Akt signaling pathway reverses sorafenib‑derived chemo‑resistance in hepatocellular carcinoma, Oncol Lett (2018). https://doi.org/10.3892/ol.2018.8536.

[7]  W. Chen, X. Liu, S. Yuan, T. Qiao, HSPA12B overexpression induces cisplatin resistance in non-small-cell lung cancer by regulating the PI3K/Akt/NF-κB signaling pathway, Oncol Lett (2018). https://doi.org/10.3892/ol.2018.7800.

[8]  H. Deng, Y. Chen, P. Li, Q. Hang, P. Zhang, Y. Jin, M. Chen, PI3K/AKT/mTOR pathway, hypoxia, and glucose metabolism: Potential targets to overcome radioresistance in small cell lung cancer, Cancer Pathogenesis and Therapy 1 (2023) 56–66. https://doi.org/10.1016/j.cpt.2022.09.001.

Reviewer 2 Report

Comments and Suggestions for Authors

The manuscript reported the biologic effects of ethanol extract of Adlay Hulls on Acute Myeloid Leukemia Cell Proliferation and found it suppressed PI3K/Akt Pathway activity. There are a few issues to be addressed. The types of active compounds that have been identified are not novel among other medicinal plants. What are the significances of the findings? In order to demonstrate its inhibitory effects on cell growth, the experiments should include alternative approach to provide supportive evidences. Among the phytochemicals identified, what are the major components that play the crucial role in the cellular activities. Finally, more updated references must be cited.

Comments on the Quality of English Language

Needs improvement and editing.

Author Response

Major concerns:

  1. The types of active compounds that have been identified are not novel among other medicinal plants. What are the significances of the findings?

Response:

Thank you for your thoughtful comment. While the types of active compounds identified in our study may not be structurally novel across all medicinal plants, the novelty and significance of this work lie in the fact that this is, to our knowledge, the first systematic investigation of anti-AML bioactive constituents derived from adlay hulls, a part of the plant that is often discarded as agricultural waste. Previous studies on adlay have primarily focused on its seeds or bran, whereas the pharmacological potential of the hulls, particularly against hematologic malignancies, remains largely unexplored.

Importantly, this study is the first to report the anti-AML activity and underlying mechanisms of adlay hulls extract. Our study demonstrates, for the first time, that the ethyl acetate subfraction of adlay hulls (AHE-EA-C) exhibits significant anti-proliferative and pro-apoptotic effects in AML cell lines U937 and HL-60, potentially through inhibition of the PI3K/AKT pathway. These findings not only fill a gap in the current literature regarding the medicinal value of adlay hulls but also introduce a previously underutilized natural resource as a promising candidate for AML drug development.

  1. In order to demonstrate its inhibitory effects on cell growth, the experiments should include alternative approach to provide supportive evidences.

Response:

Thank you for your valuable suggestion. In response to your comment, we have included an additional EdU incorporation assay to further validate the inhibitory effects of AHE-EA-C on cell proliferation by assessing DNA synthesis in AML cells. The results clearly demonstrate a dose-dependent reduction in EdU-positive cells following AHE-EA-C treatment, indicating suppressed DNA replication. These new data have been added to the revised manuscript as Supplementary Figure S2 (As shown below). Corresponding descriptions in the Materials and Methods (lines 212–222) and Results sections (lines 270–279) have also been added and highlighted in red in the revised version. We believe this additional experiment provides strong supportive evidence for the anti-proliferative activity of AHE-EA-C.

  1. Among the phytochemicals identified, what are the major components that play the crucial role in the cellular activities.

Response:

Thank you for your insightful comment. In our study, phytochemical profiling revealed that flavonoids constitute a major proportion of the active subfraction AHE-EA-C, with representative compounds such as 5,7,4’-trihydroxyflavanone and neohesperidin. Notably, neohesperidin is a well-characterized flavonoid with reported anti-leukemic activity. Given the abundance of flavonoids in AHE-EA-C and accumulating evidence highlighting their roles in modulating cell proliferation, apoptosis, and signaling pathways in cancer, we speculate that flavonoids may serve as the principal bioactive constituents contributing to the anti-AML effects of AHE-EA-C [1–4].

  1. Finally, more updated references must be cited.

Response:

Thank you for your valuable suggestion. In response to your comment, we have carefully reviewed all references cited in the manuscript. Several outdated references that were no longer essential to the context have been removed, and more recent and relevant literature has been added to ensure the scientific rigor and timeliness of the citations. All newly added references have been highlighted in red in the revised manuscript for your convenience.

References:

[1]  H. Shi, X.-Y. Li, Y. Chen, X. Zhang, Y. Wu, Z.-X. Wang, P.-H. Chen, H.-Q. Dai, J. Feng, S. Chatterjee, Z.-J. Li, X.-W. Huang, H.-Q. Wei, J. Wang, G.-D. Lu, J. Zhou, Quercetin Induces Apoptosis via Downregulation of Vascular Endothelial Growth Factor/Akt Signaling Pathway in Acute Myeloid Leukemia Cells, Front. Pharmacol. 11 (2020) 534171. https://doi.org/10.3389/fphar.2020.534171.

[2]  A.A. Mahbub, C.L. Le Maitre, N.A. Cross, N. Jordan-Mahy, The effect of apigenin and chemotherapy combination treatments on apoptosis-related genes and proteins in acute leukaemia cell lines, Sci Rep 12 (2022) 8858. https://doi.org/10.1038/s41598-022-11441-z.

[3]  D. Zheng, Y. Zhou, Y. Liu, L. Ma, L. Meng, Molecular Mechanism Investigation on Monomer Kaempferol of the Traditional Medicine Dingqing Tablet in Promoting Apoptosis of Acute Myeloid Leukemia HL-60 Cells, Evidence-Based Complementary and Alternative Medicine 2022 (2022) 1–11. https://doi.org/10.1155/2022/8383315.

[4]  Y. Hou, X. Meng, K. Sun, M. Zhao, X. Liu, T. Yang, Z. Zhang, R. Su, Anti-cancer effects of ginsenoside CK on acute myeloid leukemia in vitro and in vivo, Heliyon 8 (2022) e12106. https://doi.org/10.1016/j.heliyon.2022.e12106.

Round 2

Reviewer 1 Report

Comments and Suggestions for Authors My suggestions have been addressed and, if necessary, resolved. I recommend to accept

Reviewer 2 Report

Comments and Suggestions for Authors

The manuscript reads better and looks fine.

Comments on the Quality of English Language

needs some editing